

# Non-invasive cortisol measurements as indicators of physiological stress responses in guinea pigs

Matthias Nemeth[1], Elisabeth Pschernig[1], Bernard Wallner[1,2] and Eva Millesi[1]

[1] Department of Behavioural Biology, University of Vienna, Vienna, Austria
[2] Department of Anthropology, University of Vienna, Vienna, Austria

Corresponding author
Matthias Nemeth,
matthias.nemeth@univie.ac.at

## ABSTRACT

Non-invasive measurements of glucocorticoid (GC) concentrations, including cortisol and corticosterone, serve as reliable indicators of adrenocortical activities and physiological stress loads in a variety of species. As an alternative to invasive analyses based on plasma, GC concentrations in saliva still represent single-point-of-time measurements, suitable for studying short-term or acute stress responses, whereas fecal GC metabolites (FGMs) reflect overall stress loads and stress responses after a species-specific time frame in the long-term. In our study species, the domestic guinea pig, GC measurements are commonly used to indicate stress responses to different environmental conditions, but the biological relevance of non-invasive measurements is widely unknown. We therefore established an experimental protocol based on the animals' natural stress responses to different environmental conditions and compared GC levels in plasma, saliva, and fecal samples during non-stressful social isolations and stressful two-hour social confrontations with unfamiliar individuals. Plasma and saliva cortisol concentrations were significantly increased directly after the social confrontations, and plasma and saliva cortisol levels were strongly correlated. This demonstrates a high biological relevance of GC measurements in saliva. FGM levels measured 20 h afterwards, representing the reported mean gut passage time based on physiological validations, revealed that the overall stress load was not affected by the confrontations, but also no relations to plasma cortisol levels were detected. We therefore measured FGMs in two-hour intervals for 24 h after another social confrontation and detected significantly increased levels after four to twelve hours, reaching peak concentrations already after six hours. Our findings confirm that non-invasive GC measurements in guinea pigs are highly biologically relevant in indicating physiological stress responses compared to circulating levels in plasma in the short- and long-term. Our approach also underlines the importance of detailed investigations on how to use and interpret non-invasive measurements, including the determination of appropriate time points for sample collections.

## INTRODUCTION

Non-invasive measurements of steroid hormones reliably reflect endocrine states and have therefore become common tools in biological and biomedical research

(*Kersey & Dehnhard, 2014*). Investigations in many species often rely exclusively on non-invasive sampling methods, because commonly applied analyses using plasma would be either not possible or simply inappropriate because their high invasiveness can cause stress responses (*Vahl et al., 2005*). Especially glucocorticoids (GCs), including cortisol and corticosterone, measured non-invasively in saliva or fecal samples can reliably indicate an individual's hypothalamic-pituitary-adrenal (HPA) axis activity and physiological stress responses (*Sheriff et al., 2011*; *Cook, 2012*). Beyond the importance of careful physiological and biological validations of non-invasive GC measurement techniques (*Goymann, 2012*; *Touma & Palme, 2005*; *Cook, 2012*), relations to the biologically active circulating concentrations in plasma should additionally be tested to determine their biological relevance. If non-invasively measured GC levels predict and resemble those in plasma, they can be used to reliably indicate the strength of physiological stress responses and related, well-described physiological actions of GCs (*Munck, Guyre & Holbrook, 1984*; *Blanchard, McKittrick & Blanchard, 2001*; *McEwen, 2008*).

GCs and other circulating steroid hormones are transported unaltered and rapidly into the salivary glands (*Riad-Fahmy et al., 1982*). Comparable to the circulating levels in plasma, saliva GC concentrations also represent single-point-of-time measurements. Such measurements are suitable for studying short-term or acute stress responses or for assessing the present physiological stress load (*Goymann, 2005*). Measurements in feces are, by contrast, not essentially affected by short-term fluctuations of circulating GC levels, because GCs are metabolized by the liver and excreted via the gut with a species-specific time lag (*Palme et al., 2005*). Fecal GC metabolite (FGM) concentrations therefore reflect long-term stress loads (*Goymann, 2005*). Accordingly, measurements in feces are usually preferred over plasma and saliva to assess an individual's overall stress load especially in wild animals (*Touma & Palme, 2005*). Although saliva cortisol and FGM levels indicate physiological stress loads in different contexts, non-invasive GC measurements correlate highly with plasma levels in various species, including horses (*Peeters et al., 2011*), cattle (*Pérez et al., 2004*), lemurs (*Cavigelli, 1999*), rats (*Thanos et al., 2009*), or humans (*Hellhammer, Wüst & Kudielka, 2009*). A biological relevance of non-invasive GC measurements is indicated if they predict plasma cortisol levels equally under basal and stimulated conditions. Moreover, the ratio of the measured concentrations in plasma and non-invasive sources must remain steady under the different conditions.

Our study species, the domestic guinea pig, is a classical laboratory model species, which is extensively used in behavioral and physiological studies, for example regarding social relationships and stress management (*Sachser, Dürschlag & Hirzel, 1998*; *Hennessy et al., 2006*), nutritional influences (*Bauer, Dittami & Huber, 2009*; *Weiler et al., 2012*), or fatty acid and lipid metabolism (*Fu & Sinclair, 2000*; *Fernandez & Volek, 2006*). Knowledge on the HPA-axis reactivity and validity of non-invasive GC measurements are crucial for several research topics conducted using this species. Non-invasive measurements of cortisol, which constitutes the major GC in guinea pigs (*Malinowska & Nathanielsz, 1974*), have been validated physiologically via ACTH-challenges (injection of adreno-corticotropic hormone to induce the adrenal glands to release GCs) for saliva cortisol levels (*Fenske, 1996*; *Fenske, 1997*) and FGMs (*Bauer et al., 2008*). Under the stressful
conditions of the ACTH-challenge, a strong correlation between GC levels in plasma and saliva was found (*Fenske, 1996*), and a gut passage time with peak concentrations of FGMs18–20 h post injection was reported (*Bauer et al., 2008*). Nonetheless, the power of the biological relevance of non-invasive GC measurements under different conditions and different physiological stress loads, indicated by circulating cortisol levels in plasma, remains unclear.

Changes in environmental conditions can evoke physiological stress responses and increased GC secretion rates (*Wingfield et al., 1998*; *Romero, 2002*). Especially changes in the social environment can strongly affect physiological stress loads and can easily be achieved via social confrontations with unfamiliar conspecifics (*Goymann et al., 1999*; *Franceschini et al., 2007*; *Kuo, Jong & Lai, 2011*). In order to demonstrate the biological relevance of non-invasive GC measurements in guinea pigs, we analyzed and compared GC levels in plasma, saliva, and fecal samples, as well as their ratios and relations during social isolations, a non-stressful situation (*Machatschke et al., 2011*; *Nemeth et al., 2015*), and after two-hour social confrontations with unfamiliar individuals in a novel environment, which also constitutes a highly stressful situation in our study species (*Wallner & Dittami, 2003*; *Nemeth et al., 2014*). We hypothesize that non-invasive GC measurements reflect plasma GC levels under basal and stressful conditions in guinea pigs. We expect elevated plasma and saliva GC levels shortly after social confrontations compared to the social isolation phase and, considering the gut passage time, a delayed increase in FGM concentrations in response to the social stressor.

## METHODS

### Animals and housing conditions

A total of 30 sexually intact male domestic guinea pigs (*Cavia aperea f. porcellus*), bred at the Department of Behavioural Biology, University of Vienna, Austria, were used in this study. The animals were $18.1 \pm 3.6$ (mean $\pm$ SD) months old and weighed $992.8 \pm 88.3$ g (mean $\pm$ SD). All animals could be individually identified by natural fur marks. Prior to the experiments, animals were kept in socially established single-sexed groups of ten individuals. Each groups' enclosure (2 m $\times$ 1.6 m) was environmentally enriched and the floor covered with standard woodchip bedding material. The daily provided food consisted of guinea pig pellets (ssniff V2233, ssniff Spezialdiäten GmbH, Soest, Germany) and some hay to promote tooth abrasion. Tap water was provided in several drinking bottles. Pellets and water were always available *ad libitum*. Animals were kept under constant conditions, including a light–dark cycle of 12/12 h, with lights on at 07:00 am, a temperature of $22 \pm 2$ °C, and a humidity of $50 \pm 5\%$.

Housing conditions and the experimental procedure were in line with EU Directive 2010/63/EU and the Austrian laws for animal experiments and animal keeping. The study was checked and approved by the internal board on animal ethics and experimentation of the Faculty of Life Sciences, University of Vienna, Austria (# 2014-005), and permitted by the Austrian Federal Ministry of Science and Research (BMWF-66.006/0024-II/3b/2013).

## Experimental procedure

The study consisted of two experiments, which were carried out on a total of 30 animals (20 animals for experiment 1 and ten animals for experiment 2). Experiment 1 was conducted to analyze and compare cortisol concentrations in plasma, saliva, and fecal samples, as well as their ratios and relations repeatedly during social isolation and after social confrontations with unfamiliar individuals in a novel environment. This approach reliably elicits physiological stress responses in all involved individuals, as it represents an equally challenging situation for all animals with no advantages of being a resource holder or of already established social hierarchies between single individuals, which can affect GC levels as well. The situation is therefore unbiased by previous behavioral and environmental factors and usually constitutes a strong stressor in all involved animals (*Wallner & Dittami, 2003*; *Nemeth et al., 2014*), while social isolations can be considered as a non-stressful situation in this species (*Machatschke et al., 2011*; *Nemeth et al., 2015*). As physiological stress loads were not reflected in feces in the first experiment, another experiment was performed to monitor FGM levels at regular intervals during social isolation and after a social confrontation. This enabled monitoring the diurnal variation, the gut passage time, and peak concentrations in response to the social stressor. Both experiments, including the different conditions (social isolation and social confrontations), were carried out in a separate test room under the same ambient conditions. For the experiments, animals were transferred from their social groups to single cages, so they were visually and socially isolated from each other. Although auditory and olfactory contact with the other isolated animals in the test room was still possible, social isolations prevented direct social interactions between the individuals, which usually results in low physiological stress loads (*Nemeth et al., 2014*). The floor of each cage (100 cm × 60 cm × 45 cm) was covered with woodchip bedding material and a shelter was provided. Guinea pig pellets and water were available *ad libitum* throughout the experiments, and a daily ration of 5 g of hay was provided to each animal.

### Experiment 1

Experiment 1 was carried out in two consecutive runs, each starting at 12:00 pm, with ten animals being tested per run. After the animals were transferred singly to cages, they were left undisturbed for the first 24 h. On the following three days (days 'Iso1,' 'Iso2,' 'Iso3'), animals remained socially isolated in their cages and baseline blood, saliva, and fecal samples were collected during this social isolation period. On each of the next three days (days 'Soc1,' 'Soc2,' 'Soc3'), all animals of the current run were socially confronted with each other at 10:00 am. Here, they were transferred to a squared arena (1.6 m × 1.6 m) placed in the same room. The walls of the arena were built of laminated fiberboard and the floor was covered with woodchip bedding material. Animals remained together within the arena until 12:00 pm for social interactions. After these daily two-hour social confrontations, they were returned to their cages. Blood, saliva, and fecal samples were collected after social confrontations in order to measure cortisol concentrations due to the stressful condition. To factor out the circadian rhythm of circulating cortisol levels (*Garris, 1979*; *Fujieda et al., 1982*), blood and saliva samples were always collected at 12:00

pm, directly after each other, during social isolation and after social confrontations. The whole procedure lasted less than three minutes per animal. Fecal samples were collected daily at 08:00 am, starting 20 h after the first collection of blood and saliva samples (referred to as '+20 h' samples in the following). This time span was previously shown to reflect peak concentrations of fecal cortisol metabolites in guinea pigs under stressful conditions using an ACTH-challenge test (*Bauer et al., 2008*). After the last collection of fecal samples, animals were returned to their single-sexed groups.

### Experiment 2

According to the procedure in experiment 1, experiment 2 animals were transferred singly to cages at 12:00 pm and left undisturbed for the first 24 h. For the next 24 h, animals remained in their cages and baseline saliva and fecal samples were collected during isolation. A single saliva sample was collected at 12:00 pm, fecal samples were collected at 12:00 pm and at 18:00 pm on the same day, and at 06:00 am on the following day (these saliva and fecal samples are referred to as 'Base' samples in the following). Afterwards, at 10:00 am, all animals were transferred to a squared arena for two hours, as described in experiment 1, and then returned to their cages. After the social confrontation, saliva samples were collected at 12:00 pm (referred to as 'postSoc' sample in the following) as a control for the acute HPA-axis reactivity to the social stressor. Starting at 14:00 pm, fecal samples were collected in two-hour intervals until 12:00 pm on the following day, 24 h after the end of the social confrontation (referred to as samples 'postSoc2,' 'postSoc4,' ..., 'postSoc24' in the following). Each sample included all the feces that were defecated during the preceding two hours. After the last collection of fecal samples (24 h after the end of the social confrontation) saliva samples were collected again (referred to as 'postSoc24' sample in the following). Throughout experiment 2, a red light was placed in the test room during the night time (19:00 pm until 07:00 am), which could not been perceived by the animals but allowed an unrestricted collection of fecal samples. After the final fecal and saliva samples, animals were returned to their single-sexed groups.

## Sample collections

Saliva samples were collected following *Fenske (1997)* by inserting a standard cotton bud into the animal's cheek pouch for 30 s. Cotton buds with saliva were stored in 1.5 ml tubes containing previously prepared brackets made of 1 ml pipette tips: a tip was cut into four pieces of the same length; the two outer pieces were discarded; the two inner pieces were slotted together and put into the tube so that the conical inner piece of the former tip could hold the cotton bud. Saliva was extracted from the cotton buds by centrifugation (10 min, 14,000 rpm, 17,968 × g). Brackets and dry cotton buds were removed and pure saliva was stored at −20 °C until further analysis.

Blood samples were collected via ear veins based on *Sachser & Pröve (1984)* with few modifications. Immediately after gently squeezing one of the ears to accumulate blood in the veins, a prominent ear vein was punctured with a sterile lancet and approximately 300 µl of the flowing blood was collected with heparinized (5,000 units) micropipettes and released into 1.5 ml tubes. Plasma was separated by centrifugation (10 min, 14,000 rpm,
17,968 × g) in two consecutive steps, to obtain pure plasma, which was then transferred to a new 1.5 ml tube and stored at −20 °C until further analysis.

Fecal samples were collected directly from the bedding material. Only those samples were collected that were definitely not contaminated with urine. Fecal samples were stored at −20 °C until further analysis.

## Hormone analyses

All hormonal analyses, using biotin-strepdavidin enzyme-linked immunoassays, and preceding hormone extractions from plasma and fecal samples followed *Palme & Möstl (1993)* and *Palme & Möstl (1997)*. Enzymes and antibodies were purchased from the University of Veterinary Medicine, Vienna, Austria. All analyses were run in duplicates; the confidence criterion for the samples was set at a coefficient of variance (CV) of ≤15% for duplicates. The CV for sample duplicates was calculated as the percentage of the standard deviation of the duplicates on the mean of the duplicates. This serves as a control of the sample and analysis quality (in CVs > 15%, samples would have to be reanalyzed or excluded from further analyses).

Cortisol was extracted from plasma samples by adding 2 ml diethyl ether (100%) to 100 μl of plasma. After vortexing the samples four times for 15 min each, with 15 min intermissions in between, they were frozen overnight at −20 °C and evaporated at 30 °C afterwards. Samples were then diluted 1:400 and analyzed using a cortisol-specific antibody (*Palme & Möstl, 1997*). Analysis of saliva cortisol concentrations did not require any preceding extraction procedures. Saliva was diluted 1:50 and analyzed using the same cortisol-specific antibody as used for plasma cortisol concentrations. Fecal samples were dehumidified at 60 °C, crushed, and 0.1 g of the samples was suspended in 2 ml methanol (80%). After centrifugation (15 min, 3,000 rpm, 825 × g), samples were diluted 1:20 and analyzed using an 11-oxoetiocholanolone antibody measuring FGMs (*Palme & Möstl, 1997*). Intra- and inter-CVs based on pooled control samples were as follow: plasma 6.9% (intra) and 4.2% (inter), saliva 11.3% (intra) and 8.6% (inter), feces 5.3% (intra) and 7.5% (inter).

## Calculations of plasma cortisol:saliva cortisol and plasma cortisol:FGM ratios

To further assess of the biological relevance of non-invasive GC measurements in experiment 1, the plasma cortisol:saliva cortisol ratio and the plasma cortisol:FGM ratio were calculated for each single animal and sampling point in time. Cortisol levels in plasma served as reference values because they represent the actual and biologically relevant stress load, but especially GC measurements in feces appeared to be higher in approximately half of the samplings. The calculations were therefore performed by using a formula, which linearizes the ratios across positive and negative values:

$$x = e^{|\ln(\frac{a}{b})|} * \frac{a-b}{|a-b|}.$$

By exponentiating Euler's number with the absolute value of the natural logarithm of the quotient of two values (in this case: $a =$ plasma cortisol levels, $b =$ saliva cortisol

or FGM levels), the absolute positive factorial difference between the two values was calculated, meaning that $x > 1$ for each calculation. Multiplication with the quotient of the difference between the two values $a$ and $b$, and the absolute value of the difference, resulted in a positive value ($x > 1$) if $a > b$ or a negative value ($x < 1$) if $a < b$, given that $a \neq b$. Considering two positive and unequal values for $a$ and $b$, dividing $a$ by $b$ will yield a value of $+x$, while dividing $b$ by $a$ will yield a value of $-x$; the mean of $a/b$ and $b/a$ would be $0$. In the present data, in which ratios were mostly undirected throughout the sampling points in time, this formula was most qualified for further statistical analyses of the ratios.

## Statistical analysis

All statistics were performed using the statistical package R version 3.1.0 (*R Development Core Team, 2013*) and additional packages 'nlme' (*Pinheiro et al., 2013*), to perform linear mixed models (LMEs) due to repeated measurement analyses, and 'phia' (*De Rosario-Martinez, 2013*) for post-hoc interaction analyses of significant effects.

Plasma, saliva, and FGM concentrations, as well as the plasma cortisol:saliva cortisol ratio and the plasma cortisol:FGM ratio in experiment 1 were analyzed separately by performing LMEs on the time course across the experiment with 'days' ('day1', 'day2', 'day3'), 'condition' ('isolation', 'social confrontation'), and their interaction as fixed effects and individual ID as random effect to correct for the repeated measurements. To determine the degree to which saliva cortisol and FGM concentrations predict plasma cortisol levels under non-stressed and stressful conditions, an LME was performed with 'saliva cortisol', 'fecal cortisol', and 'condition' ('isolation', 'social confrontation'), including interactions of condition with saliva cortisol and fecal cortisol as fixed effects, and individual ID as random effect. To assess the immunoassay's quality, an LME was performed on the sample duplicates' CVs, including the 'sampling type' ('plasma', 'saliva', 'feces') and the 'actual cortisol concentration' as fixed effects, and individual ID and sampling type as random effect, to control for repeated measures within single individuals and across the different sampling types. Significant factorial effects were further analyzed by applying post-hoc interaction analyses with Bonferroni corrections.

Saliva cortisol and FGM concentrations in experiment 2 were analyzed by applying LMEs with 'sampling time' (saliva cortisol: 'Base', 'postSoc', 'postSoc24'; fecal cortisol: 'Base', 'postSoc2', 'postSoc4', …, 'postSoc24') as fixed factor and individual ID as random effect due to the repeated measurements. Another LME was calculated with saliva cortisol as response variable, FGM levels as fixed effect, and individual ID as random effect, to analyze whether fecal cortisol levels predict saliva cortisol levels.

Model assumptions (linearity, normal distribution, and homogeneity of variance) were checked by visually inspecting of residual and fitted value plots as well as by performing Shapiro–Wilk normality tests and Levene's tests for homogeneity of variance. Some of the data had to be transformed by applying the natural logarithm or by taking the second or third root. To ease interpretation, non-transformed data were used for visualizations whenever this was appropriate. Significance was set at a level of $p \leq 0.05$. All values represent means $\pm$ standard errors of the mean.

## RESULTS

### Experiment 1

#### Plasma cortisol, saliva cortisol, and FGM concentrations

Plasma cortisol concentrations differed significantly between and within the experimental conditions (days: $F_{2,95} = 0.370$, $p = 0.692$; condition: $F_{1,95} = 62.234$, $p < 0.001$; days:condition: $F_{2,95} = 5.077$, $p = 0.008$) (Fig. 1A) and were generally higher after social confrontations compared to the social isolation period. While no differences were detected between the single days of social isolation ($\chi^2 = 0.739$, $p = 1$), plasma cortisol concentrations were significantly higher after the first social confrontation compared to the third (day1–day2: $\chi^2 = 2.670$, $p = 0.614$; day1–day3: $\chi^2 = 16.854$, $p < 0.001$; day2–day3: $\chi^2 = 6.108$, $p = 0.081$).

Saliva cortisol concentrations were exclusively affected by the experimental conditions, with no day-related effects (days: $F_{2,95} = 0.872$, $p = 0.422$; condition: $F_{1,95} = 19.401$, $p < 0.001$; days:condition: $F_{2,95} = 1.750$, $p = 0.179$) (Fig. 1B). Corresponding to plasma cortisol, saliva cortisol levels were significantly increased after social confrontations.

Concentrations of FGMs measured in the +20 h samples showed significant differences between and within the experimental conditions (days: $F_{2,95} = 3.671$, $p = 0.029$; condition: $F_{1,95} = 9.454$, $p = 0.003$; days:condition: $F_{2,95} = 3.541$, $p = 0.033$) (Fig. 1C) and were generally lower during the three days of social confrontations compared to the social isolation period. While an almost significant difference was detected between the single days throughout the social isolation period ($\chi^2 = 7.342$, $p = 0.051$), where animals showed slightly increased levels on the first day, no differences between the single days were detected after social confrontations ($\chi^2 = 1.779$, $p = 0.822$).

Only saliva cortisol concentrations proved successful in predicting plasma cortisol levels ($F_{1,98} = 78.154$, $p < 0.001$; $R^2 = 0.402$) (Fig. 2), which was not the case for FGM levels ($F_{1,98} = 2.128$, $p = 0.148$; $R^2 < 0.001$). These effects did not differ between experimental conditions (saliva:condition $F_{1,95} = 1.599$, $p = 0.209$, feces:condition $F_{1,95} = 3.705$, $p = 0.057$).

#### Plasma cortisol:saliva cortisol and plasma cortisol:FGM ratios

The logarithmized ratio of plasma and saliva cortisol showed no significant changes throughout the experiment (days: $F_{2,95} = 0.531$, $p = 0.590$; condition: $F_{1,95} = 0.620$, $p = 0.433$; days:condition: $F_{2,95} = 0.491$, $p = 0.613$). The mean logarithmized ratio was 2.328 (9.944 after back transformation).

For the logarithmized ratio of plasma cortisol and FGM levels, significant differences were found between and within the experimental conditions (days: $F_{2,95} = 2.920$, $p = 0.059$; condition: $F_{1,95} = 51.880$, $p < 0.001$; days:condition: $F_{2,95} = 6.148$, $p = 0.003$), with values <0 during isolation and >0 during social confrontations. No differences were determined between the single days during the social isolation period ($\chi^2 = 5.840$, $p = 0.108$), but a significantly higher ratio was detected after the first social confrontation compared to the third (day1–day2: $\chi^2 = 0.890$, $p = 1$; day1–day3: $\chi^2 = 8.016$, $p = 0.028$; day2–day3: $\chi^2 = 3.563$, $p = 0.354$).

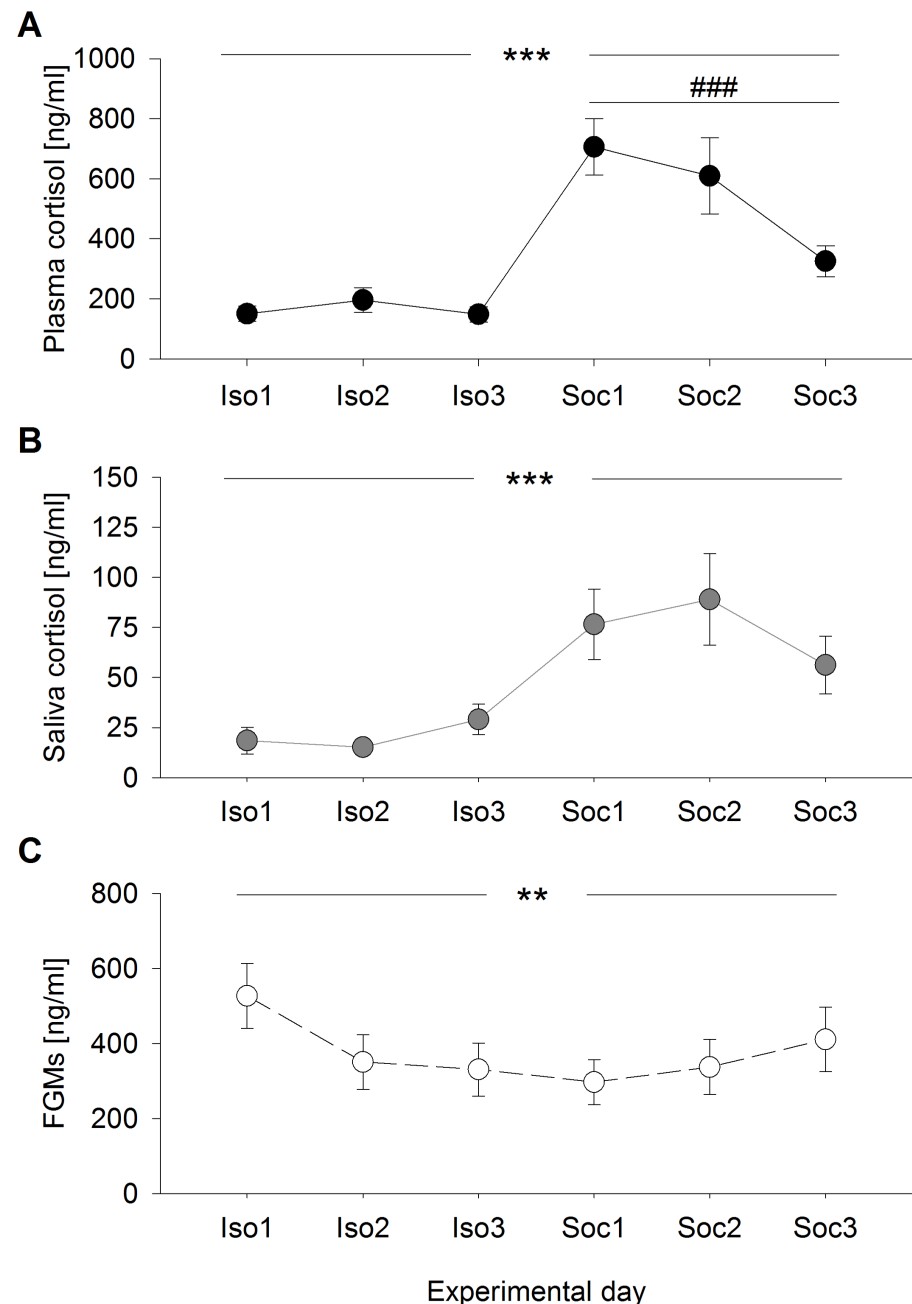

**Figure 1** **Glucocorticoid levels during social isolation and after social confrontations in experiment 1.**
(A) Plasma cortisol concentrations, (B) saliva cortisol concentrations, and (C) fecal glucocorticoid
metabolite (FGM) concentrations (+20 h samples as control of the overall stress load) on six consecutive
days, including three days of social isolation (Iso1, Iso2, Iso3) and three days of two-hour social
confrontations (Soc1, Soc2, Soc3). **$p \leq 0.01$, ***$p \leq 0.001$ comparing both experimental conditions
(social isolation and social confrontation); ### $p \leq 0.001$ comparing experimental days Soc1 and Soc3.

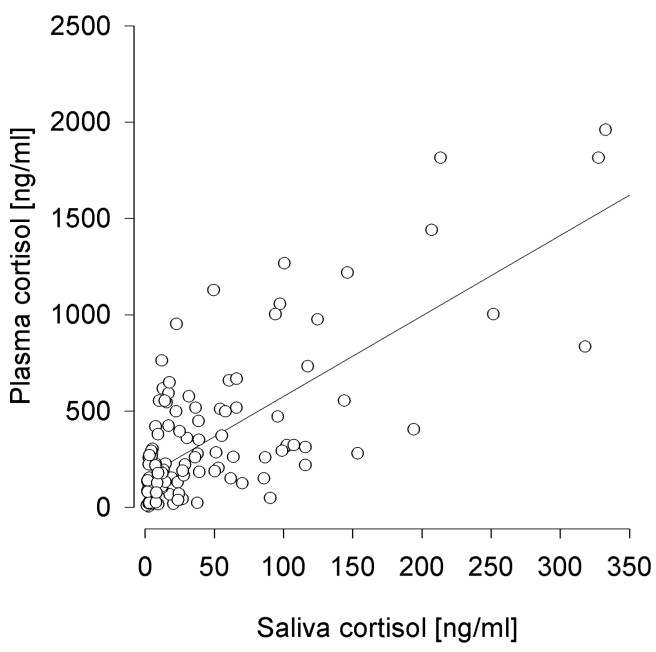

**Figure 2** **Prediction of plasma cortisol concentrations by saliva cortisol concentrations.** $R^2 = 0.402$.

### Coefficients of variance (CV) for duplicates of plasma cortisol, saliva cortisol, and FGM levels

The CVs for duplicate sample analyses, serving as a quality control for the hormonal analyses, differed significantly between the sampling types but showed no relations to the actual cortisol concentrations (sample: $F_{2,38} = 7.021$, $p < 0.001$; cortisol: $F_{1,297} = 2.564$, $p = 0.110$; sample:cortisol: $F_{2,297} = 1.702$, $p = 0.184$). The mean CV was significantly lowest for FGMs, highest for saliva cortisol, and intermediate for plasma cortisol (plasma–saliva: $\chi^2 = 8.193$, $p = 0.013$; plasma–feces: $\chi^2 = 6.480$, $p = 0.033$; saliva–feces: $\chi^2 = 15.718$, $p < 0.001$) (Fig. 3).

### Experiment 2
#### Saliva cortisol concentrations

Saliva cortisol concentrations measured throughout experiment 2 showed significant differences across the sampling points ($F_{2,18} = 49.142$, $p < 0.001$). Highest levels were measured directly after the social confrontation, with significantly lower levels 24 h before and 24 h after social confrontation (Base: $12.34 \pm 1.88$ ng/ml, postSoc: $71.85 \pm 14.07$ ng/ml, postSoc24: $6.55 \pm 1.01$ ng/ml; Base-postSoc: $\chi^2 = 49.704$, $p < 0.001$; Base-postSoc24: $\chi^2 = 6.306$, $p = 0.036$; postSoc-postSoc24: $\chi^2 = 91.416$, $p < 0.001$).

#### FGM concentrations

Baseline FGM concentrations measured three times within 24 h before the social confrontation at 12:00 pm, 18:00 pm, and 06:00 am did not differ from each other ($F_{2,18} = 1.584$, $p = 0.235$; 12:00 pm: $319.25 \pm 54.71$ ng/g, 18:00 pm: $316.82 \pm 41.07$ ng/g, 06:00 am: $411.64 \pm 83.12$ ng/g). The mean of these samples was therefore calculated and

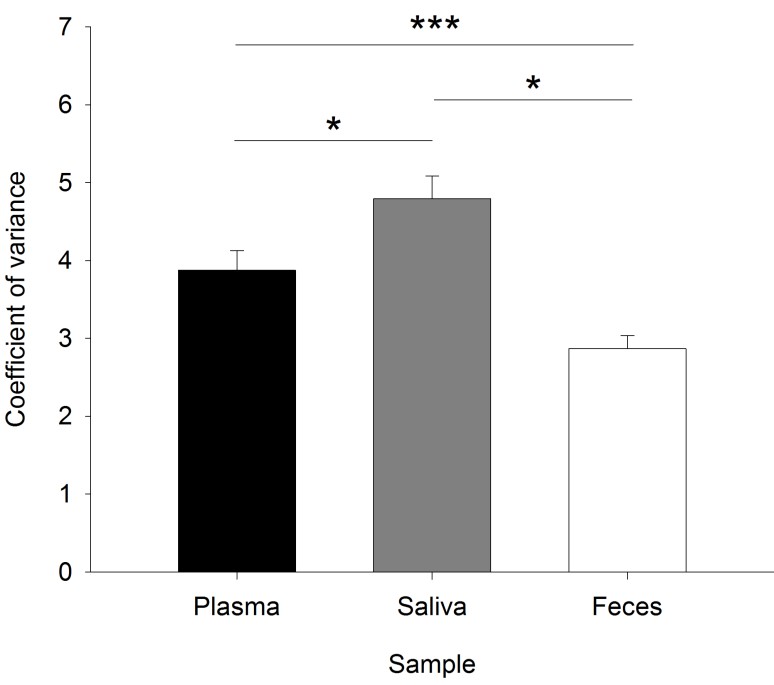

**Figure 3 Coefficients of variance for the duplicate measurements of glucocorticoids in plasma, saliva, and fecal samples as a control of the sample and analysis quality.** The coefficient of variance is the percentage of the standard deviation of the duplicates on the mean of the duplicates. $*p \leq 0.05$, $***p \leq 0.001$.

served as an overall baseline value for the analysis of FGM levels measured in two-hour intervals after the social confrontation.

The time course of FGM levels, including the mean baseline value measured on the day before the social confrontation and values measured in two-hour intervals until 24 h after the social confrontation, was highly significant ($F_{12,108} = 13.8247, p < 0.001$) (Fig. 4). Significantly increased FGM levels were detected from four to twelve hours after the end of the social confrontation compared to mean baseline values, with peak concentrations occurring six hours after the social confrontation (2 h: $t = 1.642, p = 0.104$; 4 h: $t = 4.439$, $p < 0.001$; 6 h: $t = 8.619, p < 0.001$; 8 h: $t = 5.972, p < 0.001$; 10 h: $t = 3.535, p = 0.001$; 12 h: $t = 2.698, p = 0.008$; 14h: $t = 0.258, p = 0.797$; 16 h: $t = 0.176, p = 0.861$; 18 h: $t = 1.507, p = 0.135$; 20 h: $t = 1.015, p = 0.312$; 22 h: $t = 1.102, p = 0.273$; 24 h: $t = 0.395$, $p = 0.693$).

It also proved successful to predict saliva cortisol concentrations by FGM levels measured 6 h afterwards ($F_{1,9} = 13.473, p = 0.005; R^2 = 0.428$) (Fig. 5).

## DISCUSSION

The present study revealed a high biological relevance of non-invasive GC measurements in saliva and fecal samples of domestic guinea pigs. Saliva cortisol and FGM levels measured in samples adjusted to the appropriate gut passage time were both significantly increased in response to the social confrontations and were highly correlated to the actual circulating

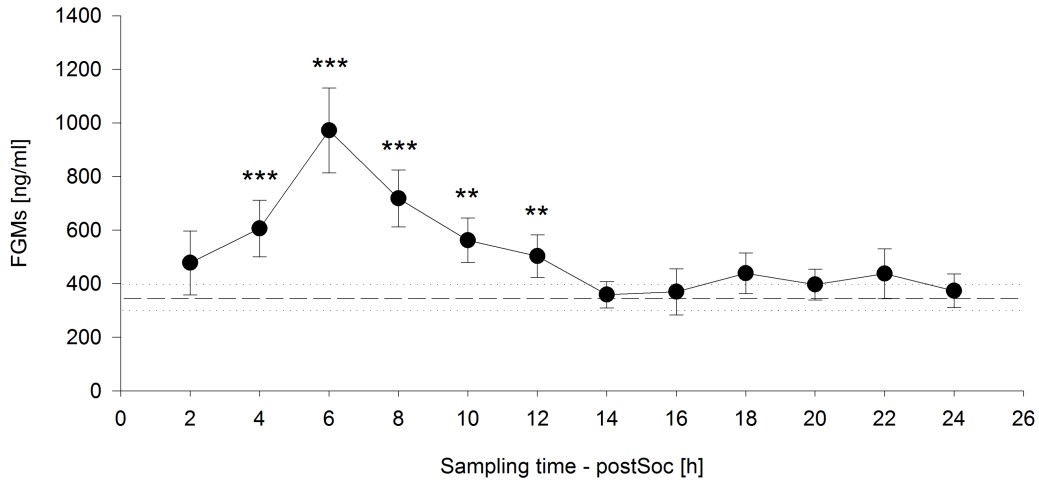

**Figure 4  Fecal glucocorticoid metabolites (FGMs) measured in two-hour intervals after social confrontation (postSoc).** The dashed line represents the mean baseline value (dotted lines: ±standard error of the mean) based on three measurements on the day before the social confrontation. **$p \leq 0.01$, ***$p \leq 0.001$ compared to baseline FGM measurements.

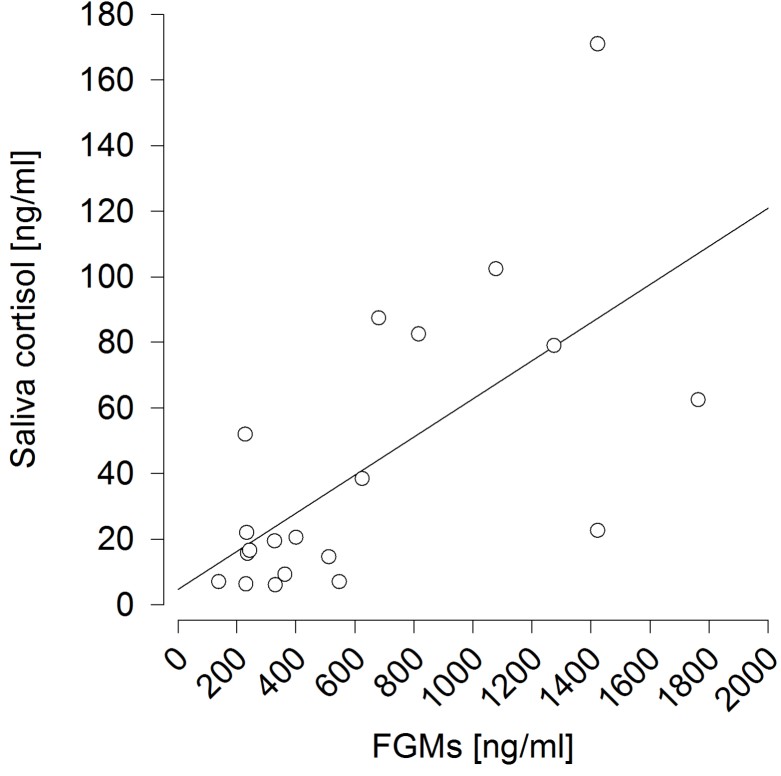

**Figure 5  Prediction of saliva cortisol concentrations by fecal glucocorticoid metabolite (FGM) concentrations measured six hours afterwards.** $R^2 = 0.428$.

cortisol levels in plasma. In addition to previously conducted physiological validation studies, our findings indicate that non-invasive GC measurements go beyond indicating the presence of a physiological stress response to also showing the strength of such a response to a prevalent stressor.

Subjecting guinea pigs to social confrontations with unfamiliar individuals in a novel environment for two hours significantly increased plasma and saliva cortisol concentrations in experiment 1. Similar responses were measured on all three days of social confrontations compared to the isolation period. This confirms our experimental approach of reliably evoking physiological stress responses by subjecting the animals to this challenging situation, as shown previously (*Nemeth et al., 2014*), without any pharmacological stimuli such as ACTH injections. FGM levels measured in the +20 h fecal samples throughout experiment 1, however, remained relatively constant. This indicates that the overall stress load in experiment 1 was not affected by the two-hour social confrontations, otherwise elevated levels throughout the period with daily applied social stress would be expected, but also that physiological stress loads are not reflected in feces 20 h after a stressful situation. Importantly, as +20 h FGM levels were not affected, we conclude that plasma and saliva cortisol levels both reflected the acute stress response to the applied social stressor and that the animals were not long-lastingly stressed. This is of major interest because numerous guinea pig studies investigate short-term or acute stress responses due to different types of stressors or environmental conditions, and these have mainly involved plasma cortisol measurements (*Machatschke et al., 2004*; *Zipser, Kaiser & Sachser, 2013*). Plasma cortisol levels could be highly predicted by saliva cortisol measurements in this study, supporting previous findings based on physiological validations in guinea pigs (*Fenske, 1996*; *Fenske, 1997*). Our results further confirm the use of saliva samples for measuring physiological stress loads because the same relations were found during non-stressful and stressful conditions. We therefore suggest that GC measurements in saliva can be reliably used in future studies to assess short-term physiological stress responses in guinea pigs. As saliva can be easily sampled with much less disturbance than blood sampling procedures, repeated saliva GC measurements can also be used to detect long-term stress loads (*Nemeth et al., 2014*). As the saliva sampling procedure lasts approximately one minute in guinea pigs, a time span in which the HPA-axis reactivity plays no role (*Reeder & Kramer, 2005*), the physiological stress load remains unaffected. Other sampling procedures, such as catheters, may also overcome possible handling effects, but their higher degree of invasiveness may be undesired in different research fields. Non-invasive or at least "mildly invasive" methods are therefore preferred and have also been developed in guinea pigs (*Bauer et al., 2008*; *Keckeis et al., 2012*).

Dissociations between plasma and saliva GC levels can be explained by the conversion to cortisone by the salivary glands and natural fluctuations (*Hellhammer, Wüst & Kudielka, 2009*). Concentrations in plasma are therefore usually ten times higher than in saliva. That ratio was also found here: the mean plasma cortisol:saliva cortisol ratio was 9.944 after back transformation. Moreover, the ratio did not change throughout the experiment and remained constant during non-stressful and stressful conditions, further underlining the strong correlation between plasma and saliva GC levels. GC concentrations in saliva

represent the free and active form only, in contrast to plasma where most GCs are bound to globulines and are therefore inactive. Accordingly, measuring GC concentrations in saliva was therefore even suggested to constitute a more reliable assessment of an individual's stress load than measurements in plasma (*Levine et al., 2007*; *Hellhammer, Wüst & Kudielka, 2009*). Although saliva sampling may be problematic in some animals (*Sheriff et al., 2011*), but the same clearly holds true for blood sampling procedures, the high biological relevance of saliva GC measurements would definitely argue for the non-invasive method.

The applied social confrontation approach in experiment 2 also demonstrated the biological relevance of GC measurements in fecal samples to assess physiological stress responses in guinea pigs. FGM levels peaked six hours after the end of the two-hour social confrontation and these measurements showed high relations to saliva cortisol concentrations measured six hours earlier, directly after the social confrontation. Baseline measurements from the day before show that FGM levels were not elevated at this time point. The detected peak therefore clearly represents the physiological stress response to the social confrontation. FGM levels were also monitored in experiment 1, but these +20 h measurements failed to predict plasma cortisol levels throughout this experiment. As previously validated physiologically via an ACTH-challenge, a 20-hour time span constituted the mean gut passage time for GCs in guinea pigs, with peak concentrations and maximum excretion rates of GCs in feces after the induced stress (*Bauer et al., 2008*). Nonetheless, no measurable responses to the repeated social confrontations or relations to circulating cortisol concentrations could be detected in the +20 h FGM measurements. The results of experiment 2 rather suggest a much faster mean gut passage time, as FGM peak levels were detected already six hours after the end, respectively eight hours after the onset of the social stressor. This time lag would correspond to more recent findings in which the gut passage time for injected radioactively marked cortisol was eight hours in guinea pigs (*Keckeis et al., 2012*), although these measurements were carried out under non-stressful conditions. *Bauer et al. (2008)* also reported FGM levels to be significantly increased already eight hours after ACTH-injections, although peak concentrations occurred much later at 18 h post injection, potentially reflecting the heavy physiological intervention caused by the ACTH-challenge.

FGMs do not represent single-point-of-time measurement, but display the integrated physiological stress load and GC secretion rates over a certain time span (*Dehnhard et al., 2003*; *Goymann, 2005*). Heavier stressors and related physiological stress responses can lead to longer-lasting excretions of FGMs in feces and/or higher FGM peak levels after a certain time span, as shown in rats by applying ACTH-challenges (*Touma, Palme & Sachser, 2004*; *Lepschy et al., 2007*). Different experimental validation procedures generally seem to lead to several possible interpretations of mean gut passage times and the time span to maximum GC excretion rates. Studies in rats, for example, revealed FGM peak concentrations of 14.8 h (*Lepschy et al., 2007*) and 16.7 h (*Bamberg, Palme & Meingassner, 2001*) after injection of radioactive corticosterone. Monitoring circadian patterns of FGM levels, however, suggested a much faster gut passage time of approximately six to nine hours (*Cavigelli et al., 2005*). Different experimental approaches may yield different physiological

stress responses, which may be indicated by dissociations in the mean gut passage times and/or peak concentrations of FGMs. Especially ACTH-challenges may constitute much more challenging situations than biological procedures, as an ACTH-challenge acts directly on the HPA-axis and does not have to be perceived as a stressor initially (*Smith et al., 2012*). As physiological stress loads might influence gut functions (*Trevisi et al., 2007*), also the mean gut passage time for FGMs may be affected. Considering these findings, the mean gut passage time for GC metabolites in guinea pigs can be assumed to be six to eight hours, as indicated by the findings of *Keckeis et al. (2012)* and experiment 2 of this study, rather than the reported 18 to 20 h by *Bauer et al. (2008)*. Misinterpretations of detected FGM peaks can be excluded because mammals do not show a double peak of high FGM excretion rates as birds do due to the combined excretion of urine and feces via the cloaca (*Hirschenhauser et al., 2012*). GC metabolites are more rapidly excreted via urine in birds and mammals (*Hirschenhauser et al., 2012*; *Touma et al., 2003*), but if urine contamination of fecal samples can be excluded, as in this study, then FGM peak levels in feces after a stressor reliably reflect the mean gut passage time.

Circulating GC levels fluctuate across the diurnal cycle (*Atkinson & Waddell, 1997*; *Banjanin, Kapoor & Matthews, 2004*), with peak levels usually occurring before the onset of activity, which makes it necessary to time sample collections appropriately. These diurnal fluctuations are also reflected in FGM levels of rats for example (*Touma, Palme & Sachser, 2004*; *Lepschy et al., 2007*). Findings in guinea pigs, however, remain controversial but suggest an absence of such diurnal variations (*Bauer et al., 2008*). FGM measurements in experiment 2, including baseline measurements before the social confrontation and the measurements in two-hour intervals to detect peak concentrations after the social confrontation, further indicate that guinea pigs show no diurnal variation in FGM levels. This is a strong argument for measuring physiological stress responses in fecal samples, as sample collections and GC measurements would remain unaffected by the time of day (*Bauer et al., 2008*). Rodents defecate very frequently, making knowledge on their gut passage time and on FGM diurnal cycles crucial. Compared to rodents, larger mammals usually defecate in less frequent intervals and show lengthier gut passage times (*Goymann et al., 1999*; *Wasser et al., 2000*). This may reflect the generally different metabolic rates in different-sized animals (*Hulbert et al., 2007*). We were able to collect fecal samples for each animal in each of the two-hour sampling intervals in experiment 2 of this study. Rarely did an animal defecate only once per sampling interval, defecation rates and the amounts of excreted feces were rather very high. We can therefore exclude a misinterpretation of the mean gut passage time for FGMs due to diurnal patterns in FGM levels, possible effects of physiological stress responses on gut functions, or to low defecation rates in general.

Hormonal analyses using enzyme-linked immunoassays are usually carried out in duplicates. This includes calculating a CV for each sample, as a criterion of sample and analysis quality. The confidence criterion of $\leq 15\%$ was reached for each sample, although statistical analyses revealed that the analysis of FGM levels using the 11-oxoetiocholanolone antibody worked best: the mean CV was significantly lower here than in plasma and saliva samples using the cortisol-specific antibody. This substantiates the accuracy and validity of the adapted antibody for FGM measurements (*Palme & Möstl, 1997*; *Möstl et al., 2002*;

*Palme, 2005*). Saliva cortisol measurements showed the highest CV, which may be due to possible contaminations with food items that can influence the sensitivity of the assay. However, the average CVs for all three types of measurements were far below the confidence criterion of 15%, which argues for the reliability of all measurements per se.

Although physiological validations have already been carried out in guinea pigs, the current study proved the biological relevance of these non-invasive measurements by monitoring GC levels in regard to the animal's natural responses to different environmental and social conditions. The experimental protocol revealed the strengths of non-invasive GC measurements in saliva and fecal samples but also underlined the importance of detailed investigations on their biological relevance. Especially measurements of FGM levels should be timed appropriately because gut passage times for GCs are species-specific and also seem to be dependent on the type of stressor and the corresponding stress response. By demonstrating the biological relevance of non-invasive GC measurements in saliva and fecal samples, we conclude that saliva cortisol and FGM levels go beyond indicating physiological stress responses to revealing the strength of such a response. This opens new opportunities to draw conclusions about the physiological and metabolic actions of GCs via non-invasive measurements.

### Funding
The authors declare that there was no funding for this study.

### Competing Interests
The authors declare there are no competing interests.

### Author Contributions
- Matthias Nemeth conceived and designed the experiments, performed the experiments, analyzed the data, wrote the paper, prepared figures and/or tables, reviewed drafts of the paper.
- Elisabeth Pschernig conceived and designed the experiments, performed the experiments, analyzed the data, contributed reagents/materials/analysis tools, wrote the paper, reviewed drafts of the paper.
- Bernard Wallner wrote the paper, reviewed drafts of the paper.
- Eva Millesi conceived and designed the experiments, contributed reagents/materials/-analysis tools, wrote the paper, reviewed drafts of the paper.

### Animal Ethics
The following information was supplied relating to ethical approvals (i.e., approving body and any reference numbers):

The experimental procedures were in line with EU Directive 2010/63/EU and the Austrian laws for animal experiments and animal keeping. The study was checked and approved by the internal board on animal ethics and experimentation of the Faculty of

Life Sciences, University of Vienna, Austria (# 2014-005), and permitted by the Austrian Federal Ministry of Science and Research (BMWF-66.006/0024-II/3b/2013).

## Data Availability

All relevant data are provided in Supplemental Information.

## Supplemental Information

Supplemental information for this article can be found online at http://dx.doi.org/10.7717/peerj.1590#supplemental-information.

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
