# Peer review of "Non-invasive cortisol measurements as indicators of physiological stress responses in guinea pigs"

_PeerJ, doi:10.7717/peerj.1590_

## Round 0.1 · original submission · Major Revisions

As you can see, the reviewers (particularly reviewer 2) have several issues with your manuscript. Please consider all the suggestions of both reviewers in any revised version of your manuscript.

·

Basic reporting

The manuscript by Nemeth et al describes experiments in guinea pigs investigating the effects of different environmental conditions (social isolation and social confrontation with an unfamiliar individual) on glucocorticoid (GC) concentrations measured in plasma, saliva, and fecal samples. In contrast to the numerous publications in humans where saliva steroid measurements are now used as an important diagnostic tool for the assessment of adrenocortical function, little information is available on saliva steroids in laboratory animals. Therefore, the aim of this study was to estimate whether non-invasive GC measurements in guinea pigs reflect their adrenocortical function.

Experimental design

In general, experiments are well designed and performed and the paper is well written and documented. All animal experiments were conducted in conformity with the prevailing ethical standards in line with EU and/or Austrian laws.

Validity of the findings

The findings are valid, previously unreported and interesting. Authors found increased plasma and saliva cortisol concentrations after social confrontation and a strong relation between plasma and saliva cortisol levels. However, fecal GC metabolites (FGMs) levels measured 20 hours after confrontations remained relatively constant indicating that the overall stress load in this experiment was not affected. Moreover, in a second study FGMs levels measured in two-hour intervals for 24 hours were significantly increased 4h to 12h after social confrontation, reaching peak concentrations already six hours after social confrontation. Taken together, this is an interesting paper, providing further evidence that GC measurements in saliva and fecal samples can be used as non-invasive method of monitoring changes in guinea pig adrenocortical function.

Additional comments

All in all, the experiments in the present study are carefully designed, well conducted, the manuscript well written and the data are presented in an easy to follow way. However, there are some minor points the authors should address before publishing.
The main concern is the missing control group that receive a novel environment exposure without social confrontations. It is well known that exposure to a novel environment per se is a potent stimulus for activating the hypothalamic-pituitary-adrenal (HPA) axis. However, in the present study the increase of GCs is exclusively related to the social confrontation. As animals were transferred from their home cages into a novel arena with an unfamiliar conspecific this means that animals were exposed to a novel spatial and social environment that should be described more precisely in the discussion.
Moreover, the question whether the term “socially isolated” is justified in this study remains unclear as animals were only visually and socially isolated from each other but would have still auditory as well as olfactory contact to other animals as they were housed in the same room.
Another critical point might be the low frequency of blood and saliva sampling. Like many other hormones, GCs have a circadian rhythm in many species demonstrated by frequent blood sampling studies. However, in experiment 2 authors collected fecal samples in a 2h interval indicating no diurnal variation in FGM levels in guinea pigs. Thus, it would be interesting to see whether this absence of diurnal variations found in FGM levels can be confirmed by correlating plasma and saliva GCs levels.

Minor points
In the methods some further information should be included concerning the definition and calculation of the coefficient of variation (CV). Also in the results and Fig. 4 should be mentioned how to calculate this statistical parameter and what it means.
In the discussion authors mentioned that “saliva samplings can be applied easily with only marginal anthropogenic influences”. However, authors should take into account that this procedure of sample collection as well as blood sampling via ear veins, which involves handling of animals may by itself be stressful and may confound the results. Therefore, manipulation-free sampling methods are preferential and can be achieved by the use of special remote blood sampling devices such as catheters.

Reviewer 2 ·

Basic reporting

see below.

Experimental design

see below.

Validity of the findings

see below.

Additional comments

General Comments. This study validated the use of a non-invasive technique - FGM levels to estimate the physiological impact of a biological stressor, a social confrontation, in guinea pigs. Although the techniques seemed appropriate and sound, overall, unfortunately, I found the study very wanting.
The authors lay this study out as a biological validation for the use of FGM in guinea pigs however, a validation has already been done and thus this study really found that a social confrontation can increase FGM levels in guinea pigs (it is not truly a validation of the technique) – although this is a relatively minor point.
A far bigger problem I found was that the authors seemed to misinterpret what FGMs indicate and over what time scale they are relevant (for example, they are altered by acute stressors). Further, they seemed to have a limited knowledge of the most pertinent studies and researchers in this field, and thus were unable to introduce or discuss their findings in light of the current state of the literature.
Lastly, I found the actual writing of the ms to be wanting. The introduction was too long and lacked clarity and focus. I was unsure of the exact purpose of the ms and how it would accomplish its objectives. It read more like a very, very limited review not like an introduction. Because of this I was unsure of why certain procedures and experiments were performed. Further, what and how the results linked back to the purpose and objective of the ms. Lastly, similar to the introduction the discussion was lacking reference and knowledge of the current literature and thus missed many important discussion points relevant to the data and results presented here (or that much knowledge and findings have already been shown about certain, seemingly, speculative discussion by the authors).
Overall, I must recommend that this ms is not acceptable in its current form. The ms requires a major revision – and even then its impact I feel will be small. The introduction needs to be much more concise and lead the reader easily to the objectives/hypothesis/predictions of the study – and particularly what the ‘point’ of many of the experiments or results are for. The methods and results could be shortened. The discussion also needs a major revision to actually discuss the results in light of the current state of knowledge and how the findings within the paper fit and advance the field. The ms as a whole should be shortened by at least 15-20%.

---

## Round 0.2 · accepted · Accept

Authors have improved the manuscript and it is now Acceptable.